# Polarized Light Microscopy-Based Quantification of Scleral Collagen Fiber Bundle Remodeling in the Lens-Induced Myopia Mouse Model

**DOI:** 10.3390/life15111743

**Published:** 2025-11-13

**Authors:** Yajing Yang, Shin-ichi Ikeda, Longdan Kang, Ziyan Ma, Kazuno Negishi, Kazuo Tsubota, Yohei Tomita, Kate Gettinger, Toshihide Kurihara

**Affiliations:** 1Department of Ophthalmology, Keio University School of Medicine, Tokyo 160-8582, Japan; yajing@keio.jp (Y.Y.); shin-ikeda@keio.jp (S.-i.I.); kazunonegishi@keio.jp (K.N.);; 2Laboratory of Photobiology, Keio University School of Medicine, Tokyo 160-8582, Japan; 3Laboratory of Chorioretinal Biology, Keio University School of Medicine, Tokyo 160-8582, Japan; 4Tsubota Laboratory, Inc., Tokyo 160-8582, Japan

**Keywords:** collagen fiber bundles, polarized light microscopy, picrosirius red staining, scleral remodeling, myopia

## Abstract

Scleral remodeling plays a crucial role in myopia development. Although ultrastructural collagen changes have been described, quantitative assessment at the fiber bundle level remains limited. This study quantitatively evaluated scleral collagen remodeling in a lens-induced myopia (LIM) mouse model using polarized light microscopy (PLM) combined with picrosirius red (PSR) staining. LIM was induced in C57BL/6J male mice by applying a monocular −30 D lens from postnatal week 3 to 6. Eyes wearing −30 D lenses showed myopic shifts in refraction (−8.42 ± 3.87 D vs. 4.42 ± 4.45 D; *p* = 0.002) and axial elongation (0.21 ± 0.04 mm vs. 0.18 ± 0.05 mm; *p* = 0.020) compared with contralateral controls. PLM revealed looser, more disorganized collagen bundles in myopic sclera. Quantitative analysis showed reduced bundle proportion (40.91 ± 6.58% vs. 48.36 ± 6.47%; *p* = 0.040) and mean size (147.11 ± 59.38 µm^2^ vs. 281.45 ± 101.00 µm^2^; *p* = 0.002). These results indicate that myopia development involves structural remodeling of the sclera and that PLM with PSR staining provides a practical approach for quantitative wide-field evaluation of collagen architecture in experimental myopia.

## 1. Introduction

Myopia is one of the leading causes of visual impairment worldwide, driven by progressive axial elongation. With its prevalence increasing at an alarming rate, it has become a significant public health concern [1,2]. Scleral remodeling, especially changes in the extracellular matrix (ECM), plays a critical role in this elongation process [3,4,5]. The scleral ECM is primarily composed of densely packed type I collagen fiber bundles interwoven with Proteoglycans [6]. Collagen in the sclera exhibits a hierarchical architecture, whereby triple-helical tropocollagen molecules assemble into fibrils, fibrils align to form fibers, and fibers further aggregate into higher-order fiber bundles (Figure 1). Disruption in collagen bundle organization can weaken the sclera’s biomechanical integrity and lower its stiffness [7,8,9,10]. Therefore, quantitative assessment of collagen fiber bundle structure may offer valuable insights into scleral remodeling and guide the development of new strategies for understanding and managing myopia.

Currently, various techniques such as atomic force microscopy [11], scanning electron microscopy [12,13] and transmission electron microscopy [14,15] have been utilized to investigate collagen remodeling in experimental myopia. Although these high-magnification methods can reveal details of individual fiber or fibril morphology and local alignment, they are limited in spatial scale, often capturing only small regions at the nanometer scale. To capture tissue-level collagen characteristics, polarization-sensitive optical coherence tomography (PS-OCT) [16,17] and ultrasound-based scanning acoustic microscopy [18] have been introduced, enabling the evaluation of overall collagen orientation, birefringence patterns, or regional mechanical properties across broader areas. However, these approaches do not offer direct visualization of fiber bundle structure.

To overcome this limitation, we employed polarized light microscopy (PLM) in combination with picrosirius red (PSR) staining. While PLM can detect collagen fibers through their intrinsic birefringence, this natural signal is often weak. PSR staining enhances molecular alignment within collagen, thereby amplifying birefringence and improving image contrast and sensitivity [19,20]. The intensity of birefringence reflects fiber packing density and orientation: densely packed, well-aligned bundles exhibit stronger signals, whereas loosely arranged fibers show weaker birefringence. At the optical resolution used in this study, PLM allows visualization of both higher-order assemblies of multiple aligned fibers, referred to as fiber bundles, and individual collagen fibers that appear as relatively isolated elongated structures [21,22,23].This integrated approach enables the wide-field quantitative analysis of collagen fiber organization in scleral tissue. Using this approach, we investigated structural remodeling of collagen fiber bundles in a lens-induced myopia (LIM) mouse model to better understand the biomechanical basis of axial elongation.

## 2. Materials and Methods

### 2.1. Animal Model

All experimental procedures were approved by the Animal Experimental Committee of Keio University and adhered to the Association for Research in Vision and Ophthalmology (ARVO) Statement for the Use of Animals in Ophthalmic and Vision Research.

Five wild-type C57BL/6J male mice (CLEA Japan, clea-japan.com) were used in this study. The mice were housed in one cage under standard laboratory conditions with free access to food and water, under approximately 50 lux fluorescent lighting on a 12 h light/12 h dark cycle.

Procedures involving LIM surgery and ocular measurements were performed under anesthesia. Mice were anesthetized via intraperitoneal injection of a mixture of midazolam (Sandoz K.K., Tokyo, Japan), medetomidine (Domitor^®^, Orion Corp., Espoo, Finland), and butorphanol tartrate (Meiji Seika Pharma Co., Tokyo, Japan), collectively referred to as MMB, at a dose of 0.1 mL per 10 g of body weight.

The LIM model was established following previously described protocols [24,25]. A small metal fixation post was secured to the skull using a self-curing dental adhesive to attach a spectacle frame. A −30 diopter (D) lens was mounted in front of the right eye, and an empty frame without lens was placed in front of the left eye as an internal control. The lenses and frames were inspected daily and cleaned at least twice per week to maintain optical clarity and proper alignment. Myopia was induced from postnatal week 3 to week 6.

Measurements of refraction and axial length were performed at postnatal week 3 and week 6 using an infrared photorefractor (Steinbeis Transfer Center, Plochingen, Germany) and spectral-domain optical coherence tomography (SD-OCT; Envisu R4310, Leica Microsystems, Wetzlar, Germany), respectively. After the final measurements, all mice were euthanized by cervical dislocation.

### 2.2. Specimen Staining and Image Acquisition

Immediately after euthanasia, eyeballs were enucleated, and residual orbital tissue was carefully removed. The intact eyeball was fixed in Super fix KY-500 (Kurabo Industries Ltd., Osaka, Japan) for 72 h, then embedded in paraffin, sectioned at 5 μm thickness, and stained with PSR (Polysciences, Warrington, PA, USA) according to the manufacturer’s protocol (Figure 2B). Briefly, deparaffinized and hydrated sections were immersed sequentially in Solution A (phosphomolybdic acid) for 2 min, Solution B (picrosirius red F3BA) for 60 min, and Solution C (0.1 N hydrochloric acid) for 2 min, then dehydrated, cleared, and mounted.

Images were captured using a BX53 microscope (Olympus, Tokyo, Japan) equipped with a polarizing module for birefringence imaging (Figure 2C). To ensure consistency, all optical parameters, including illumination intensity, polarization angle, exposure time, and magnification, were kept constant throughout the imaging process. As birefringence depends on the orientation and packing of collagen fibers, the microscope stage was rotated to obtain consistent orientation of the scleral tissue across all samples (Figure 3). This procedure ensured accurate visualization of birefringence and enabled more reliable interpretation of collagen organization.

### 2.3. Quantification of Posterior Scleral Collagen Structure

The scleral region was analyzed using a 40× objective lens. To ensure consistency across samples, the analysis focused on an area located 300–600 μm from the optic nerve head, corresponding to the posterior sclera [26,27]. This region was chosen specifically to avoid the peripapillary zone, where the sclera and choroid are structurally entangled and difficult to delineate. Quantitative image analysis was performed using ImageJ software (version 1.54p; National Institutes of Health, Bethesda, MD, USA) [28].

### 2.4. Proportion of Collagen Fiber Bundles

All images were first converted to 8-bit grayscale format (Figure 4A,B). Two threshold levels were applied to segment the collagen structures. The higher threshold isolated densely packed fiber bundles (Figure 4C), while the lower threshold captured all visible collagen fibers, including both bundles and finer individual strands (Figure 4D). The proportion of fiber bundles was calculated as the ratio of high-threshold to low-threshold areas.

Threshold values were initially determined using representative images based on visual discrimination between dense bundles and finer collagen strands. These fixed values were then uniformly applied to all images to maintain segmentation consistency.

### 2.5. Average Size of Collagen Fiber Bundles

In this study, the size of a collagen fiber bundle was defined as its cross-sectional area. To obtain this measurement, collagen fiber bundles were identified using the higher threshold determined in the previous step (Figure 5A). The “Analyze Particles” function in ImageJ was then applied, which uses a “connected component labeling” algorithm to distinguish adjacent pixels and group them into separate objects [29]. This process enabled the identification of discrete collagen fiber bundles with well-defined boundaries (Figure 5B). The average size was calculated as the mean area of all labeled particles.

### 2.6. Statistical Analysis

Statistical analyses were performed using GraphPad Prism software (version 9.5.1; GraphPad Software Inc., La Jolla, CA, USA). The normality of each dataset was assessed using the Shapiro–Wilk test prior to statistical comparisons. For normally distributed data, paired *t*-tests were used to evaluate differences between groups. Data are presented as mean ± standard deviation (SD), and Cohen’s dz effect sizes with 95% confidence intervals (CIs) were calculated to estimate the magnitude and precision of the effects. A *p*-value of <0.05 was considered statistically significant.

## 3. Results

### 3.1. Refraction and Axial Length Alterations in Myopia

The eyes treated with −30 D lenses exhibited a significant myopic shift in refraction (−30 D vs. no lens: −8.42 ± 3.87 D vs. 4.42 ± 4.45 D; *p* = 0.002, dz = 2.60, 95% CI: 6.70–18.97) and axial length elongation (−30 D vs. no lens: 0.21 ± 0.04 mm vs. 0.18 ± 0.05 mm; *p* = 0.020, dz = 1.34, 95% CI: −0.05 to 0.00) compared to the contralateral controls (Figure 6A,B).These consistent optical and biometric alterations indicate that the −30 D lens treatment effectively induced a reproducible myopic phenotype after three weeks of lens wear.

### 3.2. Collagen Fiber Bundle Alterations in Myopia

Quantitative analysis revealed a significant decrease in both the proportion (−30 D vs. no lens: 40.91 ± 6.58% vs. 48.36 ± 6.47%, *p* = 0.040, dz = 1.04, 95% CI: −0.01 to 0.16) and average size (−30 D vs. no lens: 147.11 ± 59.38 µm^2^ vs. 281.45 ± 101.00 µm^2^, *p* = 0.002, dz = 2.65, 95% CI: 71.36 to 197.30) of collagen fiber bundles in myopic eyes compared with the contralateral controls (Figure 6C,D).

This pattern was also visually evident in polarized light microscopy images, which showed that collagen fiber bundles in the posterior sclera of myopic eyes appeared more loosely packed and exhibited disorganized alignment compared to contralateral controls (Figure 7). These visual findings support the quantitative results, indicating that myopia induction was associated with clear microstructural disorganization of the collagen bundle network.

## 4. Discussion

In this study, we applied PLM with PSR staining to visualize collagen fiber bundles in the sclera of a LIM mouse model. We also developed and validated a novel image-based approach for quantitative assessment of collagen remodeling. Through this method, we found that both the proportion and average size of collagen fiber bundles were significantly reduced in the myopic eyes compared to the contralateral controls.

The analysis specifically focused on the posterior scleral region, as this area exhibits the most pronounced remodeling during myopia progression and is therefore regarded as a representative site for assessing structural alterations [3]. The contralateral eye was used as an internal control to minimize inter-animal variability. Although cross-ocular influences cannot be entirely ruled out, previous studies have demonstrated that scleral remodeling in myopia is largely a locally regulated process [3], with no comparable structural changes observed in the fellow eye. Our findings on collagen fiber bundle disruption in the myopic sclera are consistent with previous observations. Jin et al. demonstrated that form-deprivation-induced high myopia in guinea pigs resulted in disorganization, twisting, fragmentation, and separation of scleral collagen bundles, as visualized by hematoxylin and eosin (H&E) staining, which was associated with increased ocular extensibility [30]. A human PS-OCT study has reported reduced scleral birefringence in highly myopic eyes, which likely reflects decreased collagen packing and reduced structural anisotropy due to disorganization of collagen fibers [16]. While their studies provided valuable insights, our approach expands this understanding by enabling quantitative assessment of collagen bundle remodeling, including both proportional loss and size reduction. This quantitative framework allows for more objective and reproducible evaluation of microstructural changes during myopia progression.

To better interpret these morphological changes, it is important to consider the optical properties of collagen fibers under polarized light. When stained with PSR, type I collagen exhibits yellow-to-red birefringence against a dark background, allowing for clear visualization and precise morphometric assessment [20]. Variations in birefringence intensity primarily reflect differences in collagen packing and alignment, with more compact and uniformly oriented fibers generating stronger signals than loosely arranged networks [21,22,23]. Additionally, fibers aligned parallel to the imaging plane appear brighter than those oriented perpendicularly or obliquely [20]. Given these properties, PLM offers a reproducible and sensitive method for assessing collagen fiber organization across wide tissue sections, making it particularly suitable for evaluating the structural remodeling observed in this study.

We applied fixed grayscale thresholds during image analysis to enable reliable and reproducible quantification across samples. In pilot testing, adaptive thresholding algorithms such as Otsu’s method tended to overestimate weak birefringence regions in the myopic sclera, likely due to the low contrast between collagen bundles and background. Therefore, a single global threshold was applied consistently to maintain comparability across samples. The higher threshold delineated tightly packed collagen bundles, while the lower threshold captured the entire birefringent collagen. Although we applied fixed thresholds in our analysis, researchers may adjust these values based on their own image characteristics, as long as thresholding is applied consistently across all samples. The robustness of this approach was supported by inter-observer validation, which indicated negligible variation between independent analyses. Based on our findings, we propose that the structural alterations observed in the myopic sclera result from a multi-level remodeling process of the ECM, including collagen fiber thinning and disorganization. Our previous studies [31] have demonstrated that endoplasmic reticulum (ER) stress is activated during myopia progression, which interferes with the synthesis and assembly of collagen, resulting in thinner individual collagen fibers and a reduction in total collagen content. In addition, collagen fibers are embedded within a proteoglycan rich, gel-like matrix that resists interlamellar shearing and stabilizes the lamellar alignment [6,9,32]. During myopia progression, matrix metalloproteinase-2 (MMP-2), a key enzyme upregulated in scleral remodeling [33], promotes degradation of the extracellular matrix [34]. With progressive matrix weakening, collagen lamellae lose structural support and begin to slide relative to one another [35], leading to scleral creep and axial elongation (Appendix A) [36,37]. As structural instability and lamellar slippage accumulate, this process may compromise the integrity of collagen bundles, leading to morphological fragmentation and the formation of smaller, less organized fiber bundles, consistent with our observations in the myopic group.

It should be noted that nearly all existing experimental myopia models have been developed using male animals, including the present study. This is primarily due to methodological considerations; male mice display stable ocular growth trajectories, avoiding variability introduced by the estrous cycle. Consequently, male models have become the global standard for establishing reproducible myopia paradigms. Nevertheless, accumulating evidence suggests that sex hormones may modulate scleral ECM metabolism and influence susceptibility to myopia [38]. Investigating female models under controlled conditions will be important for elucidating sex-dependent mechanisms and improving the translational relevance of experimental myopia research.

This study has certain limitations. Since our analysis is based on two-dimensional PLM images, regions with low birefringence intensity cannot be definitively interpreted. Such weak signals may result from collagen fibers that are oriented obliquely or perpendicularly to the imaging plane, or from fibers that are parallel to the plane but loosely packed or disorganized. Therefore, our quantification primarily reflects the organization of fiber bundles aligned parallel to the imaging plane. Future studies using polarization-resolved second-harmonic generation microscopy may provide true three-dimensional insights into collagen organization beyond the two-dimensional limitations of PLM. In addition, the sample size was relatively small, which may have limited the ability to detect more subtle effects. Nonetheless, the trends observed were consistent across all samples, supporting the overall reliability of the findings. Future studies incorporating three-dimensional imaging and larger sample sizes will be important for further validation.

## 5. Conclusions

In conclusion, this study provides new insights into scleral remodeling by quantitatively demonstrating a reduction in both the proportion and size of collagen fiber bundles in the myopic sclera. These changes may weaken the biomechanical integrity of the sclera and contribute to axial elongation. By combining PSR staining with PLM, we have established a quantitative method for wide-field assessment of collagen architecture, offering a useful tool for future research on myopia progression and potential therapeutic approaches.

## Figures and Tables

**Figure 1 life-15-01743-f001:**
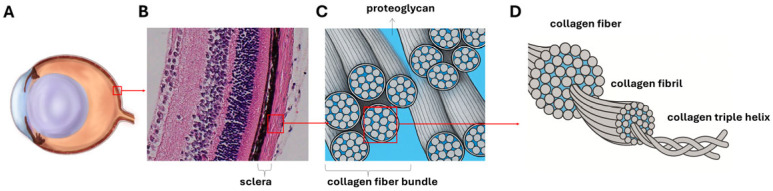
Multiscale illustration of the hierarchical organization of scleral collagen. From left to right: (**A**) A schematic diagram of the sagittal section of the mouse eyeball, with a red box marking a representative area of the ocular wall to illustrate its layered organization. (**B**) A histological section stained with hematoxylin and eosin (H&E), showing the layered structure of the ocular wall, with a red box indicating the representative region of the sclera used for schematic illustration. (**C**) A conceptual diagram illustrating the microarchitecture of the sclera, showing that collagen fiber bundles consist of densely packed collagen fibers, with a red box highlighting a representative collagen fiber. (**D**) A schematic representation of the molecular organization of collagen fibrils, which constitute the substructure of each fiber.

**Figure 2 life-15-01743-f002:**
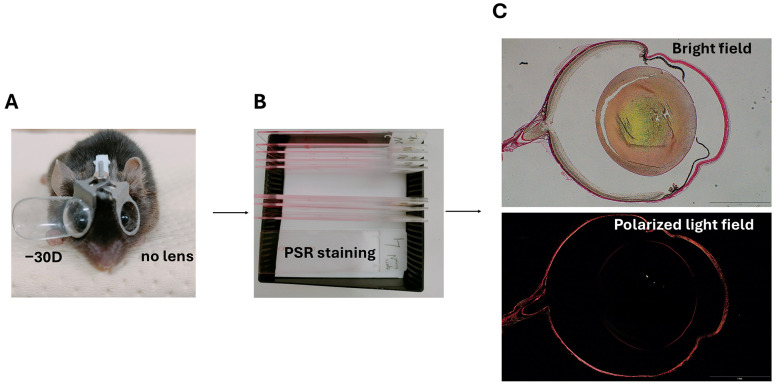
A schematic overview of the experimental workflow. (**A**) Lens-induced myopia was established in C57BL/6J male mice from 3 to 6 weeks of age using monocular −30 D lenses on the right eye, while the left eye received no lens. (**B**) After euthanasia, eyeballs were enucleated, paraffin-embedded, sectioned, and stained with picrosirius red (PSR). (**C**) Representative images of sections captured under bright field (top) and polarized light field (bottom). Collagen fiber bundles appear birefringent under polarized light, showing yellow-to-red colors characteristic of PSR staining.

**Figure 3 life-15-01743-f003:**
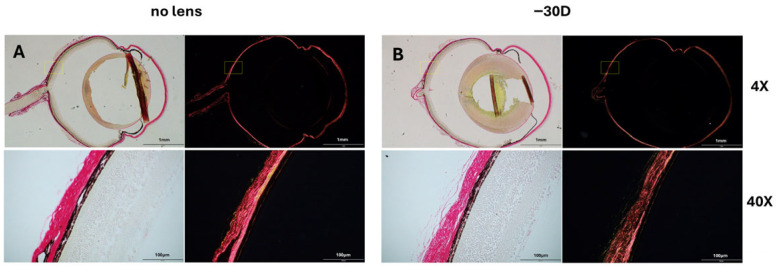
Representative PSR-stained scleral sections under bright field and polarized light field. (**A**) Control eye (no lens); (**B**) Myopic eye (−30 D lens). Images were acquired at low (4×) and high (40×) magnifications. To ensure comparability of birefringence signals, all optical settings were kept constant, and the microscope stage was rotated to align the orientation of the scleral tissue across samples. Under polarized light, collagen fibers exhibit yellow-to-red birefringence typical of PSR staining. Yellow boxes indicate the regions selected for higher-magnification imaging.

**Figure 4 life-15-01743-f004:**
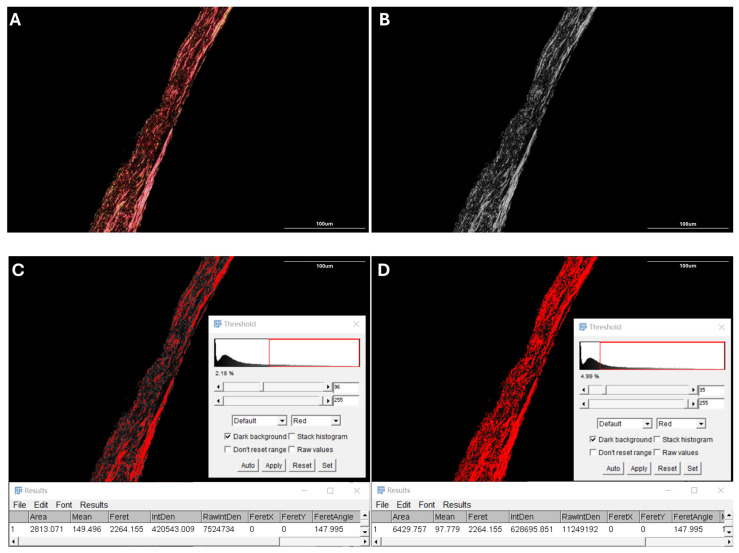
Representative image processing for collagen fiber bundle quantification. (**A**,**B**) PLM image of a PSR-stained posterior scleral section (**A**) and its grayscale-converted version (**B**) used for thresholding. (**C**,**D**) ImageJ-based segmentation using preset thresholds. The high threshold (**C**) identifies dense fiber bundles, while the low threshold (**D**) outlines the full birefringent collagen fiber network. The area ratio between the two masks was used to quantify the proportion of fiber bundles. Red areas in panels C and D indicate regions selected by ImageJ during threshold processing.

**Figure 5 life-15-01743-f005:**
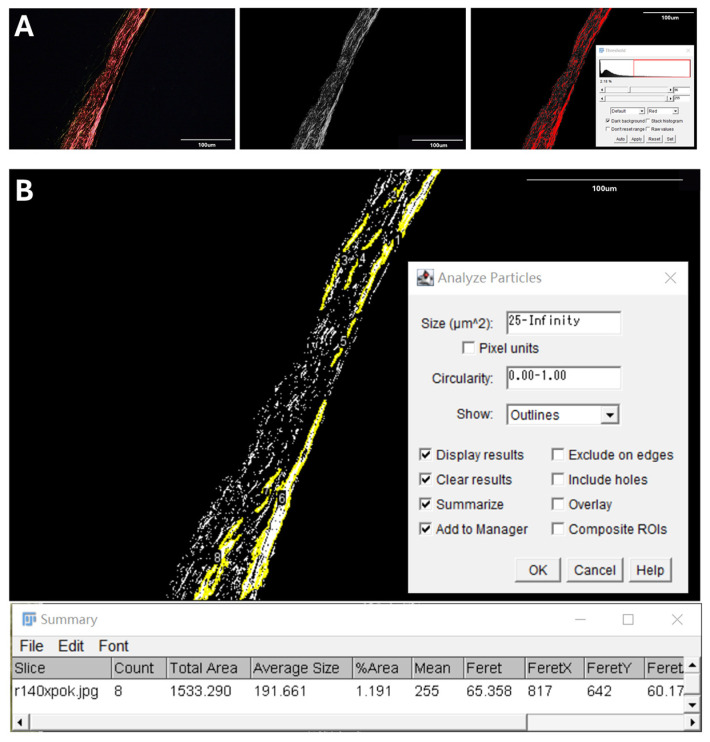
Quantitative analysis of average collagen fiber bundle size. (**A**) Image processing workflow for identifying densely packed collagen fiber bundles, including a polarized light image, its grayscale-converted version, and the thresholded image used for segmentation. (**B**) The “Analyze Particles” function in ImageJ was applied to the segmented images to measure the cross-sectional area of individual collagen fiber bundles. Yellow outlines indicate detected particles. The average size was calculated based on the total area and object count.

**Figure 6 life-15-01743-f006:**
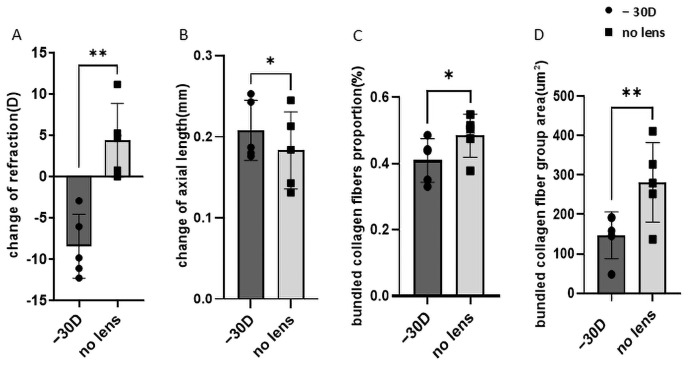
Refraction, axial length, and collagen fiber bundle alterations in lens-induced myopia. Compared to no-lens controls, −30 D eyes showed a significant myopic shift in refraction (**A**) and axial elongation (**B**). In the posterior sclera, the proportion of densely packed collagen fiber bundles was significantly lower in the −30 D group (**C**), and the average cross-sectional area of individual bundles was also markedly reduced (**D**). Data are presented as mean ± SD. *p* < 0.05 (*), *p* < 0.01 (**); paired *t*-test, *n* = 5 per group.

**Figure 7 life-15-01743-f007:**
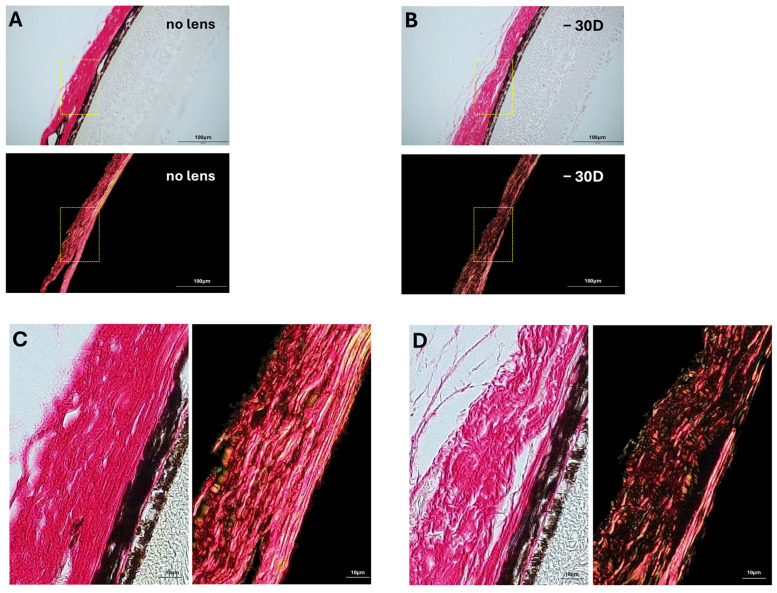
Structural differences in posterior scleral collagen fiber bundles between control and myopic eyes. (**A**,**B**) Representative PSR-stained scleral sections from control (no lens) and myopic (−30 D lens) eyes, respectively, imaged under bright field (top) and polarized light microscopy (bottom). Yellow boxes indicate regions selected for magnification. (**C**,**D**) Enlarged views of the regions highlighted in (**A**,**B**), respectively. In the control eye (**C**), collagen fiber bundles appear densely packed and regularly aligned. In contrast, the myopic sclera (**D**) shows more loosely arranged and disorganized collagen bundles. Colors represent PSR staining under bright field and polarized light.

## Data Availability

The original contributions presented in this study are included in the article/Appendix A. Further inquiries can be directed to the corresponding author.

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
