# Peer review of "Polarized Light Microscopy-Based Quantification of Scleral Collagen Fiber Bundle Remodeling in the Lens-Induced Myopia Mouse Model"

_life, 2025, doi:10.3390/life15111743_

Round 1
Reviewer 1 Report
Comments and Suggestions for Authors
The manuscript is of high quality overall, presenting well-supported results. The authors employed polarized light microscopy combined with picrosirius red staining to quantitatively analyze scleral collagen fiber bundles in a mouse model of myopia. The experimental design is appropriate and effectively captures the characteristics of structural remodeling. I have a few suggestions as follows,
- In the abstract, I would suggest to report the values for myopia group first, eg., -8.42±3.87D vs 4.42±4.45D, since the myopic shift, axial elongation, and reduced bundle proportion are from myopia group. Ensure that p-values are consistently reported with three decimal places.
- Please clarify why a model without a lens was used in the contralateral eye, not 0D. Is there any evidence showing differences or consistency between using no lens versus a 0D lens?
- More details should be added in method section. For example, what light-dark cycle were the animals maintained on? Were food and water available ad libitum? Also, the PSR staining procedure and LIM protocol should also be briefly described to facilitate reproducibility.
- Choosing an area 300-600 μm away from the optic nerve is scientifically meaningful, but it would be clearer to specify which anatomical region this corresponds to in the eye, does it represent the posterior pole? Additionally, could the authors clarify why an area between the posterior pole and peripheral sclera was not analyzed as supplementary evidence?
- The results section could be expanded to provide more detailed descriptions and interpretations of the findings, rather than focusing only on number change in the figure presentation.
- The manuscript occasionally uses the terms “fiber” and “fiber bundle.” It may be helpful to briefly define these terms at the beginning of the methods section to make clear the structural level analyzed by PLM.
- Are there any myopia studies reporting comparable results regarding collagen fiber bundles size? If so, it would be helpful to include them in discussion as supporting evidence.
Author Response
Comment 1. In the abstract, I would suggest to report the values for myopia group first, eg., -8.42±3.87D vs 4.42±4.45D, since the myopic shift, axial elongation, and reduced bundle proportion are from myopia group. Ensure that p-values are consistently reported with three decimal places.
Response:
We appreciate the reviewer’s helpful suggestion. In the revised abstract, main text, and figure, the numerical order has been changed to report the myopia group values first, emphasizing that the changes originate from the myopia group. All p-values throughout the manuscript have been consistently formatted to three decimal places.
Line numbers corresponding to the revised content:
Abstract (page 1: line 20-24), main text (page 8: line 194-205), and figure 6 (page 8: line 207).
Comment 2. Please clarify why a model without a lens was used in the contralateral eye, not 0D. Is there any evidence showing differences or consistency between using no lens versus a 0D lens?
Response:
We thank the reviewer for this insightful question. In lens-induced myopia studies, both “no-lens” and “0 D lens” setups are commonly used as internal controls. In our laboratory, we compared these two control types and found no statistically significant differences in refraction, axial length[1], choroidal thickness[2], retinal thickness and retinal function[3]. Therefore, we consider these two approaches equivalent. In this study, we selected the no-lens condition primarily for practical reasons, it simplifies daily animal care and minimizes unnecessary handling of the control side, while producing results equivalent to those obtained with a 0 D lens.
[1] Jiang X, Kurihara T, Kunimi H, et al. A highly efficient murine model of experimental myopia. Sci Rep. 2018;8(1):2026. Published 2018 Feb 1. doi:10.1038/s41598-018-20272-w
[2] Ma Z, Jeong H, Yang Y, et al. Contralateral effect in progression and recovery of lens-induced myopia in mice. Ophthalmic Physiol Opt. 2023;43(3):558-565. doi:10.1111/opo.13125
[3] Yang Y, Tomita Y, Lee D, et al. Impact of Extended Lens-Induced Myopia on Retinal Structure and Function in Mice. Curr Eye Res. Published online September 16, 2025. doi:10.1080/02713683.2025.2557590
Comment 3. More details should be added in method section. For example, what light-dark cycle were the animals maintained on? Were food and water available ad libitum? Also, the PSR staining procedure and LIM protocol should also be briefly described to facilitate reproducibility.
Response:
We appreciate the reviewer’s valuable comment. Additional methodological details have been incorporated into the revised manuscript to enhance clarity and reproducibility.
Page 3 line 83-86:
Five wild-type C57BL/6J male mice (CLEA Japan, clea-japan.com) were used in this study. The mice were housed in one cage under standard laboratory conditions with free access to food and water, under approximately 50 lux fluorescent lighting on a 12 h light/12 h dark cycle.
Page 4 line 110-113:
Briefly, deparaffinized and hydrated sections were immersed sequentially in Solution A (phosphomolybdic acid) for 2 min, Solution B (picrosirius red F3BA) for 60 min, and Solution C (0.1 N hydrochloric acid) for 2 min, then dehydrated, cleared, and mounted.
Page 3 line 92-98:
A small metal fixation post was secured to the skull using a self-curing dental adhesive to attach a spectacle frame. A −30 diopter (D) lens was mounted in front of the right eye, and an empty frame without lens was placed in front of the left eye as an internal control. The lenses and frames were inspected daily and cleaned at least twice per week to maintain optical clarity and proper alignment. Myopia was induced from postnatal week 3 to week 6.
Comment 4. Choosing an area 300-600 μm away from the optic nerve is scientifically meaningful, but it would be clearer to specify which anatomical region this corresponds to in the eye, does it represent the posterior pole? Additionally, could the authors clarify why an area between the posterior pole and peripheral sclera was not analyzed as supplementary evidence?
Response:
We sincerely thank the reviewer for this insightful comment. The analyzed area (300–600 μm away from the optic nerve head) does not correspond exactly to the posterior pole, but rather to the posterior sclera. In previous studies[1,2], the posterior sclera was defined based on anatomical landmarks and visual boundaries, referring to the region posterior to the equator and anterior to the peripapillary zone. Following this convention, the quantified zone in our study represents the mid-posterior portion of the sclera. We further quantified this area as 300–600 µm from the optic nerve head in the present study to ensure consistent angular and linear measurements and achieve standardized quantification. We focused on this region for two main reasons:(1) the posterior sclera exhibits the most prominent remodeling during myopia progression, making it the most representative site for detecting structural changes[3]; and (2) other regions such as the equatorial sclera are thinner and tend to deform during paraffin embedding and sectioning, making it difficult to maintain uniformity across all samples. Therefore, we focused on the posterior sclera as the most reliable and reproducible region to demonstrate our quantitative method.
[1] Gogola A, Jan NJ, Brazile B, et al. Spatial Patterns and Age-Related Changes of the Collagen Crimp in the Human Cornea and Sclera. Invest Ophthalmol Vis Sci. 2018;59(7):2987-2998. doi:10.1167/iovs.17-23474
[2] Jan NJ, Brazile BL, Hu D, et al. Crimp around the globe; patterns of collagen crimp across the corneoscleral shell. Exp Eye Res. 2018;172:159-170. doi:10.1016/j.exer.2018.04.003
[3] Rada JA, Shelton S, Norton TT. The sclera and myopia. Exp Eye Res. 2006;82(2):185-200. doi:10.1016/j.exer.2005.08.009
Page 5: line 139-141:
To ensure consistency across samples, the analysis focused on an area located 300–600 μm from the optic nerve head, corresponding to the posterior sclera.
Page 10: line 238-240:
The analysis specifically focused on the posterior scleral region, as this area exhibits the most pronounced remodeling during myopia progression and is therefore regarded as a representative site for assessing structural alterations.
Comment 5. The results section could be expanded to provide more detailed descriptions and interpretations of the findings, rather than focusing only on number change in the figure presentation.
Response:
We sincerely thank the reviewer for this valuable suggestion. In the revised manuscript, we have expanded the Results section by adding more detailed descriptions for each part.
Page 8: line 197-199:
These consistent optical and biometric alterations indicate that the −30 D lens treatment effectively induced a reproducible myopic phenotype after three weeks of lens wear.
Page 9: line 219-221:
These visual findings support the quantitative results, indicating that myopia induction was associated with clear microstructural disorganization of the collagen bundle network.
Comment 6. The manuscript occasionally uses the terms “fiber” and “fiber bundle.” It may be helpful to briefly define these terms at the beginning of the methods section to make clear the structural level analyzed by PLM.
Response:
We thank the reviewer for this insightful suggestion. To clarify the collagen hierarchy and the structural level, we have added a concise definition at the Introduction section. The revised text reads as follows:
Page 1-2, line 36-38:
Collagen in the sclera exhibits a hierarchical architecture, whereby triple-helical tropocollagen molecules assemble into fibrils, fibrils align to form fibers, and fibers further aggregate into higher-order fiber bundles.
Page 2-3, line 67-72:
The intensity of birefringence reflects fiber packing density and orientation: densely packed, well-aligned bundles exhibit stronger signals, whereas loosely arranged fibers show weaker birefringence. At the optical resolution used in this study, PLM allows visualization of both higher-order assemblies of multiple aligned fibers, referred to as fiber bundles, and individual collagen fibers that appear as relatively isolated elongated structures.
In addition, the previous Supplementary Figure 1 has been moved to the main Figure 1 in the revised version to help readers better understand the hierarchical organization of collagen structures.

Figure 1. Multiscale illustration of the hierarchical organization of scleral collagen.
From left to right: (A) A schematic diagram of the sagittal section of the mouse eyeball, with a red box marking a representative area of the ocular wall to illustrate its layered organization. (B) A histological section stained with hematoxylin and eosin (H&E), showing the layered structure of the ocular wall. (C) A conceptual diagram illustrating the microarchitecture of the sclera, showing that collagen fiber bundles consist of densely packed collagen fibers. (D) A schematic representation of the molecular organization of collagen fibrils, which constitute the substructure of each fiber.
Comment 7. Are there any myopia studies reporting comparable results regarding collagen fiber bundles size? If so, it would be helpful to include them in discussion as supporting evidence.
Response:
We thank the reviewer for this valuable comment. To the best of our knowledge, no previous myopia studies have quantitatively reported comparable results regarding collagen fiber bundle size. Our method is simple yet innovative, enabling direct visualization and quantification of collagen bundles. As described in the Introduction, polarization-sensitive optical coherence tomography (PS-OCT) and ultrasound-based scanning acoustic microscopy can detect alterations at the fiber bundle level in the myopic sclera, allowing evaluation of overall collagen orientation, birefringence patterns, and regional mechanical properties across broader areas. However, these techniques do not provide direct visualization of individual fiber bundles. As mentioned in the Discussion, conventional bright-field microscopy of H&E-stained sections can also visualize tissue-level collagen bundles. Yet, because all tissue components are stained and visible under bright-field illumination, this method is not suitable for precise quantification of collagen organization. Collectively, our method complements these imaging approaches by offering direct morphological visualization and quantitative assessment of collagen bundle remodeling.
Reviewer 2 Report
Comments and Suggestions for Authors
Dear editor
Thank you for providing me the opportunity to review this manuscript.
The paper is not without interest, but the study protocol does have its weak points. Here are my comments and suggestions:
1- Only 5 male mice were used, which is underpowered for detecting subtle effects or variability and severely limits statistical confidence. Variability in refraction and collagen metrics could be large among animals. No power calculation is provided. This raises concerns about type II errors and reproducibility. Suggest performing a post-hoc power analysis or justifying the n based on effect sizes from pilots. Otherwise perform the same analysis in an expanded cohort (n ≥ 10) to increase statistical power and validate reproducibility.
2- Exclusive use of male mice limits generalizability, as myopia prevalence and scleral biology may differ by sex (e.g., hormonal influences on ECM). It causes sex bias. Discuss this limitation and recommend future inclusion of females.
3- The authors used fixed threshold values based on representative images. This could introduce bias. Automatic or adaptive thresholding (e.g., Otsu’s method) could increase objectivity. Inter-observer agreement was qualitatively mentioned but not quantified (e.g., Cohen’s kappa or ICC).
4- Analysis focused on 300-600 µm from the optic nerve head to avoid peripapillary entanglement, but no rationale for why this represents global posterior sclera. Compare with other regions (e.g., equatorial sclera) or justify based on myopia literature.
5- Paired t-tests are appropriate, but effect sizes and confidence intervals should be reported. Small-sample normality assumptions should be validated (e.g., Shapiro–Wilk test) or use non-parametric alternatives if needed.
6- Contralateral eyes as controls are appropriate, but potential systemic effects (e.g., from monocular induction) are not discussed. Variability in bundle metrics (high SDs) suggests need for more replicates.
7- The manuscript mentions rotating the microscope stage to align samples, but the reproducibility of birefringence intensity calibration is not demonstrated. Include calibration using collagen-rich control tissues (e.g., rat tail tendon) imaged under identical settings to confirm optical stability.
8- No statistical links between biometric changes (refraction/axial length) and collagen metrics (e.g., Pearson's correlation). Correlate mechanical data with collagen bundle metrics to strengthen causal interpretation. This would strengthen causality claims.
9- No assessment of inter-animal variability or outlier influence is discussed.
10- The discussion proposes ER stress and MMP-2 roles but lacks direct evidence from this study, this distinction should be made explicit.
11- In discussion, avoid overstatement (e.g., line 229: "Ultimately disrupts" implies causality without direct proof). Compare with human studies more (e.g., ref. 16 on PS-OCT in high myopia).
12- The limitation of 2D imaging is acknowledged, but potential solutions (e.g., PLM tomography or second-harmonic generation microscopy) could be mentioned.
13- Figures 5C–D present mean ± SD values but lack individual data points, making it difficult to judge variability. Include scatter plots of individual animal data. Report effect size and confidence intervals, not just p-values, to better represent statistical reliability.
Author Response
1- Only 5 male mice were used, which is underpowered for detecting subtle effects or variability and severely limits statistical confidence. Variability in refraction and collagen metrics could be large among animals. No power calculation is provided. This raises concerns about type II errors and reproducibility. Suggest performing a post-hoc power analysis or justifying the n based on effect sizes from pilots. Otherwise perform the same analysis in an expanded cohort (n ≥ 10) to increase statistical power and validate reproducibility.
Response:
We thank the reviewer for this valuable and constructive comment. We acknowledge that the relatively small sample size (n = 5) may limit the ability to detect subtle effects. To quantitatively assess this, we conducted post-hoc power analyses based on the observed effect sizes for each parameter:
- Refraction: dz = 2.60, α = 0.05, n = 5 pairs → power > 0.99
- Axial length: dz = 1.34 → power = 0.79
- Collagen fiber bundles Proportion: dz = 1.04 → power = 0.61
- Collagen fiber bundles Average size: dz = 2.65 → power > 0.99
These analyses indicate moderate to high statistical power for most parameters (0.79–>0.99), while the collagen percentage showed relatively limited power (0.61), but all samples showed a consistent decrease, and the difference reached statistical significance. This consistent trend may indicate a biologically relevant change, but it should be interpreted with caution and warrant validation in a larger cohort.
We agree that expanding the cohort size would further enhance statistical confidence; however, due to the long experimental period and logistical constraints inherent to animal studies, additional experiments could not be conducted within the current revision timeline. Nevertheless, this limitation has been acknowledged in the revised Discussion.
Page 11-12: line 310-314:
In addition, the sample size was relatively small, which may have limited the ability to detect more subtle effects. Nonetheless, the trends observed were consistent across all samples, supporting the overall reliability of the findings. Future studies incorporating three-dimensional imaging and larger sample sizes will be important for further validation.
2- Exclusive use of male mice limits generalizability, as myopia prevalence and scleral biology may differ by sex (e.g., hormonal influences on ECM). It causes sex bias. Discuss this limitation and recommend future inclusion of females.
Response:
We sincerely thank the reviewer for raising this valuable and scientifically important point.
Sex differences represent a critical but often underappreciated factor in myopia research. In accordance with the reviewer’s suggestion, we have added a discussion on this issue in the revised Discussion section.
Worldwide, most experimental myopia models have been established using male animals. This choice is primarily driven by scientific and experimental control considerations. Female mice undergo regular estrous cycles characterized by marked fluctuations in estrogen and progesterone levels. Such hormonal variability can increase data noise and experimental uncertainty, thereby reducing reproducibility. In contrast, male mice show more linear and stable ocular growth patterns, providing statistical advantages for standardized model development.
Nevertheless, we fully agree with the reviewer that using only males is a limitation. Previous studies have suggested that sex hormones may regulate scleral ECM metabolism[1], and therefore, exclusive use of males could mask sex-specific differences.
In our own laboratory, we performed lens-induced myopia (LIM) experiments in both male and female mice starting at 3 weeks of age. However, the female mice did not show a statistically significant axial elongation after three weeks of induction (Response Figure1)[2], indicating a lower sensitivity to lens-induced myopic stimulus under the same conditions. For this reason, male mice were selected in the present study to ensure model stability and reproducibility.
We are aware that an increasing number of studies are now reintroducing female models to explore sex-dependent mechanisms. Our group is also actively optimizing conditions to establish a stable female model. This direction is clinically meaningful, as large epidemiological studies have shown that adolescent females tend to develop myopia earlier and at a higher prevalence than males[3]. Thus, incorporating female animals in future experiments would help to better simulate sex-related differences in myopia progression, scleral remodeling, and treatment responses, thereby improving the translational relevance of experimental myopia research.

Response Figure 1. Changes in axial length in female mice after 3 weeks of −30 D lens-induced myopia. No significant differences were observed between the -30D and control eyes. (Adapted from Figure 1 in reference [2].)
Page 11: line 294-302:
It should be noted that nearly all existing experimental myopia models have been developed using male animals, including the present study. This is primarily due to methodological considerations: male mice display linear and stable ocular growth trajectories, avoiding variability introduced by the estrous cycle. Consequently, male models have become the global standard for establishing reproducible myopia paradigms. Nevertheless, accumulating evidence suggests that sex hormones may modulate scleral ECM metabolism and influence susceptibility to myopia. Investigating female models under controlled conditions will be important for elucidating sex-dependent mechanisms and improving the translational relevance of experimental myopia research.
[1] Huang L, Zhang D, Zhou J. Myopia development: multifactorial interplay, molecular mechanisms and possible strategies. Front Med (Lausanne). 2025;12:1638184. Published 2025 Aug 26. doi:10.3389/fmed.2025.1638184
[2] Zhang Y, Mori K, Jeong H, et al. Myopic shift in female mice after ovariectomy. Sci Rep. 2024;14(1):22946. Published 2024 Oct 3. doi:10.1038/s41598-024-74337-0
[3] Machluf Y, Israeli A, Cohen E, Chaiter Y, Mezer E. Dissecting the complex sex-based associations of myopia with height and weight. Eye (Lond). 2024;38(8):1485-1495. doi:10.1038/s41433-024-02931-7
3- The authors used fixed threshold values based on representative images. This could introduce bias. Automatic or adaptive thresholding (e.g., Otsu’s method) could increase objectivity. Inter-observer agreement was qualitatively mentioned but not quantified (e.g., Cohen’s kappa or ICC).
Response:
We sincerely thank the reviewer for this valuable comment regarding image quantification. Threshold selection indeed affects the objectivity and reproducibility of the results; therefore, we have added corresponding clarifications in the revised manuscript.
1.Rationale for using a global fixed threshold
In this study, a single global threshold was applied consistently across all images. The threshold was determined from representative control images and overall grayscale histograms under identical imaging and staining conditions, ensuring comparability among samples.
Although adaptive thresholding algorithms (e.g., Otsu’s method) were tested, the relatively weak birefringence signals and low contrast of collagen fiber bundles in myopic sclera caused these methods to automatically lower the threshold in dim regions, thereby overestimating collagen-positive areas[1]. Under uniform illumination and staining, a global fixed threshold provided higher stability and consistency for cross-sample comparisons[2].
2. Validation of threshold robustness and inter-observer consistency
In practice, two independent observers independently set threshold values that were almost identical under the same imaging conditions, indicating negligible inter-observer variation.
To further evaluate robustness, we intentionally introduced a ±5% threshold variation as an exaggerated scenario, far exceeding the actual inter-observer difference, to test whether such perturbation would affect quantitative outcomes.
The intraclass correlation coefficients (ICC, two-way mixed model, absolute agreement) among the original, +5%, and −5% thresholds were: No-lens group: ICC = 0.83, indicating good consistency; −30D group: ICC = 0.96, indicating excellent consistency. These results demonstrate that collagen quantification remains reliable and reproducible even under extreme threshold variations.
3.Recalculated validation using shifted thresholds
We further reanalyzed the collagen fiber ratio and area using the new ±5% thresholds. The results showed consistent trends with the original analysis: Lowered threshold (−5%): collagen ratio P = 0.0873, collagen area P = 0.0031; Raised threshold (+5%): collagen ratio P = 0.0254, collagen area P = 0.0232.
When the thresholds were shifted by ±5%, the recalculated collagen ratio and area showed highly consistent trends with the original analysis, indicating that the statistical outcomes were robust to small threshold variations.
Page 10: line 269-273:
In pilot testing, adaptive thresholding algorithms such as Otsu’s method tended to overestimate weak birefringence regions in the myopic sclera, likely due to the low contrast between collagen bundles and background. Therefore, a single global threshold was applied consistently to maintain comparability across samples.
[1] Sezgin, M., & Sankur, B. (2004). Survey over image thresholding techniques and quantitative performance evaluation. Journal of Electronic Imaging, 13(1), 146-168.
[2] Greiner C, Grainger S, Farrow S, et al. Robust quantitative assessment of collagen fibers with picrosirius red stain and linearly polarized light as demonstrated on atherosclerotic plaque samples. PLoS One. 2021;16(3):e0248068. Published 2021 Mar 18. doi:10.1371/journal.pone.0248068
4- Analysis focused on 300-600 µm from the optic nerve head to avoid peripapillary entanglement, but no rationale for why this represents global posterior sclera. Compare with other regions (e.g., equatorial sclera) or justify based on myopia literature.
Response:
We sincerely thank the reviewer for this insightful comment.
In a previous studies[1,2], the posterior sclera was defined based on anatomical landmarks and visual boundaries, referring to the region posterior to the equator and anterior to the peripapillary zone. Following this convention, we further quantified this area as 300–600 µm from the optic nerve head to ensure consistent angular and linear measurements under identical magnification.
The lower bound (300 µm) was set to avoid the peripapillary zone where collagen fibers merge radially with the optic nerve sheath. The upper bound (600 µm) corresponded to the edge of the 40× field of view in our standardized imaging setup, providing a fixed frame and ensuring cross-sample comparability. Although extending the analysis slightly further (e.g., to 800 µm) would still fall within the posterior sclera anatomically, doing so would increase curvature variability and sectioning artifacts, making it difficult to maintain geometric uniformity among samples.
We selected this region for two additional reasons: (1) the posterior sclera exhibits the most prominent remodeling during myopia progression, making it the most representative site for detecting structural changes[3]; and (2) more peripheral regions such as the equatorial sclera are thinner and prone to bending or deformation during paraffin embedding and sectioning.
Therefore, the 300–600 µm from the optic nerve head was chosen as the most anatomically relevant and technically reproducible region for quantifying posterior scleral remodeling.
Page 5: line 139-141:
To ensure consistency across samples, the analysis focused on an area located 300–600 μm from the optic nerve head, corresponding to the posterior sclera.
Page 10: line 238-240:
The analysis specifically focused on the posterior scleral region, as this area exhibits the most pronounced remodeling during myopia progression and is therefore regarded as a representative site for assessing structural alterations.
[1] Gogola A, Jan NJ, Brazile B, et al. Spatial Patterns and Age-Related Changes of the Collagen Crimp in the Human Cornea and Sclera. Invest Ophthalmol Vis Sci. 2018;59(7):2987-2998. doi:10.1167/iovs.17-23474
[2] Jan NJ, Brazile BL, Hu D, et al. Crimp around the globe; patterns of collagen crimp across the corneoscleral shell. Exp Eye Res. 2018;172:159-170. doi:10.1016/j.exer.2018.04.003
[3] Rada JA, Shelton S, Norton TT. The sclera and myopia. Exp Eye Res. 2006;82(2):185-200. doi:10.1016/j.exer.2005.08.009
5- Paired t-tests are appropriate, but effect sizes and confidence intervals should be reported. Small-sample normality assumptions should be validated (e.g., Shapiro–Wilk test) or use non-parametric alternatives if needed.
Response:
We sincerely thank the reviewer for this important comment.
We initially used the non-parametric Wilcoxon signed-rank test, the non-parametric counterpart of the paired t-test.
Following the reviewer’s valuable suggestion, we have now validated the normality of all parameters (refraction, axial length, collagen fiber bundles Proportion, and collagen fiber bundles Average size) using the Shapiro–Wilk test.
All datasets satisfied the normality assumption (P > 0.05), and therefore paired t-tests were applied for group comparisons.
We acknowledge that the previous version did not explicitly describe this statistical validation, which has now been added to the Methods section to enhance the statistical rigor and transparency of the study.
Page 8: line 193-197:
The eyes treated with −30 D lenses exhibited a significant myopic shift in refraction (−30 D vs. no lens: -8.42 ± 3.87D vs. 4.42 ± 4.45 D; P = 0.002, dz = 2.60, 95% CI: 6.70–18.97) and axial length elongation (−30 D vs. no lens: 0.21 ± 0.04 mm vs. 0.18 ± 0.05 mm; P = 0.020, dz = 1.34, 95% CI: -0.05 to 0.00) compared to the contralateral controls.
Page 8: line 202-206:
Quantitative analysis revealed a significant decrease in both the proportion (−30 D vs. no lens: 40.91 ± 6.58% vs. 48.36 ± 6.47%, P = 0.040, dz = 1.04, 95% CI: -0.01 to 0.16) and average size (−30 D vs. no lens: 147.11 ± 59.38µm2 vs. 281.45 ± 101.00 µm2, P = 0.002, dz = 2.65, 95% CI: 71.36 to 197.30) of collagen fiber bundles in myopic eyes compared with the contralateral controls.
6- Contralateral eyes as controls are appropriate, but potential systemic effects (e.g., from monocular induction) are not discussed. Variability in bundle metrics (high SDs) suggests need for more replicates.
Response:
We sincerely thank the reviewer for this valuable comment.
We have carefully addressed both the potential systemic effects associated with the use of contralateral eyes as controls and the issue of variability in bundle metrics.
In myopia research, unilateral lens-induced myopia (LIM) with the fellow eye serving as an internal control is a widely adopted experimental design. This approach effectively minimizes inter-animal variability, such as differences in ocular size, scleral thickness, and genetic background, thereby improving the comparability and statistical robustness of the results.
We also fully acknowledge the limitations of this model. Visual signaling, neurotransmitter levels, or humoral factors may interact between the two eyes to some extent. However, previous evidence suggests that scleral remodeling in myopia is largely a locally regulated process, and although systemic mechanisms may exist, there is no evidence showing comparable morphological changes in the fellow eye[1]. Based on these findings, we consider the use of contralateral eyes as internal controls to remain scientifically justified.
We have added a brief clarification in the Discussion to acknowledge this potential cross-ocular influence and to cite supporting evidence from previous studies.
As noted in our response above, we also acknowledge that the relatively small sample size (n = 5) may limit the ability to detect subtle effects. To quantitatively assess this limitation, we conducted post-hoc power analyses based on the observed effect sizes for each parameter: • Refraction: dz = 2.60, α = 0.05, n = 5 pairs → power > 0.99; • Axial length: dz = 1.34 → power = 0.79; • Collagen percentage: dz = 1.04 → power = 0.61; • Collagen area: dz = 2.65 → power > 0.99.
These analyses indicate moderate to high statistical power for most parameters (0.79 – > 0.99), whereas the collagen percentage showed limited power (0.61), suggesting that the current sample size may be underpowered to fully capture subtle variability in this metric. Nevertheless, all samples exhibited a consistent decreasing trend, which supports the biological plausibility of this change, though the result should be interpreted with caution and validated in a larger cohort. We agree that increasing the number of replicates would further strengthen statistical confidence; however, due to the long experimental cycle and practical constraints inherent to animal studies, additional experiments could not be completed at this stage.
Page 10: line 240-244:
The contralateral eye was used as an internal control to minimize inter-animal variability. Although cross-ocular influences cannot be entirely ruled out, previous studies have demonstrated that scleral remodeling in myopia is largely a locally regulated process, with no comparable structural changes observed in the fellow eye.
[1] Rada JA, Shelton S, Norton TT. The sclera and myopia. Exp Eye Res. 2006;82(2):185-200. doi:10.1016/j.exer.2005.08.009
7- The manuscript mentions rotating the microscope stage to align samples, but the reproducibility of birefringence intensity calibration is not demonstrated. Include calibration using collagen-rich control tissues (e.g., rat tail tendon) imaged under identical settings to confirm optical stability.
Response:
We thank the reviewer for raising this important comment regarding the reproducibility of polarization light microscopy (PLM) imaging. We understand that the reviewer wishes to confirm whether rotating the microscope stage could affect the consistency of birefringence intensity measurements.
In this study, stage rotation was performed solely to align the principal orientation of the sample (scleral collagen fiber direction) parallel to the polarization axis, allowing for comparable orientation imaging. This operation does not alter the illumination angle, the orientation of the polarizer and analyzer, or the exposure settings. All images were acquired under identical conditions, fixed illumination intensity, polarization angle, exposure time, and magnification, and were completed consecutively within a single imaging session.
Under these conditions, the PLM system demonstrated excellent optical stability. We have added these technical details in the revised Methods section to clarify the reproducibility and reliability of PLM imaging.
Page 4: line 124-126:
To ensure consistency, all optical parameters, including illumination intensity, polarization angle, exposure time, and magnification, were kept constant throughout the imaging process.
8- No statistical links between biometric changes (refraction/axial length) and collagen metrics (e.g., Pearson's correlation). Correlate mechanical data with collagen bundle metrics to strengthen causal interpretation. This would strengthen causality claims.
Response:
We sincerely thank the reviewer for this constructive suggestion.
As changes in axial length directly reflect scleral elongation, we performed correlation analyses between axial elongation and collagen bundle parameters.
The results showed a moderate negative correlation between collagen bundle proportion and axial elongation (r = –0.61, p = 0.061), and a similar trend between collagen bundle area and axial elongation (r = –0.54, p = 0.11).
Although the correlations did not reach statistical significance due to the limited sample size, both parameters exhibited negative trends. This direction of these changes aligns with the expected biological relationship between scleral collagen remodeling and myopia progression, supporting the proposed structural–functional association.
These results have been verified but were not included in the revised manuscript to maintain conciseness.
9- No assessment of inter-animal variability or outlier influence is discussed.
Response:
We thank the reviewer for raising this important point regarding inter-animal variability and potential outlier influence.
To evaluate data consistency, all datasets were examined using the interquartile range (IQR) boxplot method (Q1 − 1.5×IQR, Q3 + 1.5×IQR).
-Refraction and axial length:
All data points fell within the normal IQR limits, and no outliers were detected in either group.
-Collagen fiber bundles Proportion:
In the no lens group, Q1 = 0.475 and Q3 = 0.515 (IQR = 0.040), yielding a normal range of 0.415–0.575. One mildly lower value (0.3776) was slightly below this threshold but still above the 3×IQR extreme cutoff (0.355), representing a mild biological variation rather than a statistical outlier. In the −30D group (Q1 = 0.350, Q3 = 0.442, IQR = 0.092), all values fell within the normal range (0.212–0.580), indicating no outliers.
-Collagen fiber bundles average size:
For the no lens group, Q1 = 252.18 and Q3 = 327.20 (IQR = 75.02), giving a range of 139.65–439.73; one mildly lower value (136.88) was slightly below this limit. For the −30D group, Q1 = 145.31 and Q3 = 191.66 (IQR = 46.35), corresponding to a range of 75.79–261.18; one lower value (47.52) fell below this limit. However, both values remained above the 3×IQR extreme cutoffs (27.12 and 10.26, respectively) and followed the same decreasing trend as the group means.
These were therefore classified as mild outliers reflecting natural biological variation rather than technical artifacts and were retained for analysis. Across all parameters, the within-group variability remained within a normal biological range. The consistent directional trends among individual animals confirm that the findings were not driven by isolated outliers, supporting the robustness of the results.
10- The discussion proposes ER stress and MMP-2 roles but lacks direct evidence from this study, this distinction should be made explicit.
Response:
We appreciate the reviewer’s insightful comment regarding the mechanistic interpretation.
In fact, the involvement of ER stress and MMP-2 in scleral remodeling has been directly demonstrated in our previous studies conducted by our co-first author, Shin-ichi Ikeda.
Specifically, we confirmed that ER stress is activated during myopia progression in the equivalent LIM model, leading to impaired collagen synthesis and fiber assembly (Response Figure 1) [1]. Furthermore, we demonstrated that pharmacological induction of ER stress upregulates MMP-2 expression in the sclera, establishing a functional link between these two pathways (Response Figure 2) [2].
We acknowledge that the previous version of the Discussion may have caused misunderstanding, and we have now clearly stated in the revised manuscript that these mechanisms were verified in our previous studies.

Response Figure 1. Transmission electron microscopy images of no lens-wearing (left panels) and minus 30 D lens-wearing (right panels) sclera. The observation revealed that the rough ER was dilated in scleral fibroblasts in LIM mice(Adapted from Figure 1 in reference [1].)

Response Figure 2 Induction of endoplasmic reticulum (ER) stress by Tunicamycin (Tm) significantly increased MMP-2 expression in the sclera (Adapted from Figure 6 in reference [2].)
Page 11: line 281-288:
Our previous studies have demonstrated that endoplasmic reticulum (ER) stress is activated during myopia progression, which interferes with the synthesis and assembly of collagen, resulting in thinner individual collagen fibers and a reduction in total collagen content. In addition, collagen fibers are embedded within a proteoglycan rich, gel-like matrix that resists interlamellar shearing and stabilizes the lamellar alignment. During myopia progression, matrix metalloproteinase-2 (MMP-2), a key enzyme upregulated in scleral remodeling [33], promotes degradation of the extracellular matrix.
[1] Ikeda SI, Kurihara T, Jiang X, et al. Scleral PERK and ATF6 as targets of myopic axial elongation of mouse eyes. Nat Commun. 2022;13(1):5859. Published 2022 Oct 10. doi:10.1038/s41467-022-33605-1
[2] Kang L, Ikeda SI, Yang Y, et al. Establishment of a novel ER-stress induced myopia model in mice. Eye Vis (Lond). 2023;10(1):44. Published 2023 Nov 1. doi:10.1186/s40662-023-00361-2
11- In discussion, avoid overstatement (e.g., line 229: "Ultimately disrupts" implies causality without direct proof). Compare with human studies more (e.g., ref. 16 on PS-OCT in high myopia).
Response:
Response Figure 2 Induction of endoplasmic reticulum (ER) stress by Tunicamycin (Tm) significantly increased MMP-2 expression in the sclera (Adapted from Figure 6 in reference [2].)
Page 11: line 281-288:
Our previous studies have demonstrated that endoplasmic reticulum (ER) stress is activated during myopia progression, which interferes with the synthesis and assembly of collagen, resulting in thinner individual collagen fibers and a reduction in total collagen content. In addition, collagen fibers are embedded within a proteoglycan rich, gel-like matrix that resists interlamellar shearing and stabilizes the lamellar alignment. During myopia progression, matrix metalloproteinase-2 (MMP-2), a key enzyme upregulated in scleral remodeling [33], promotes degradation of the extracellular matrix.
[1] Ikeda SI, Kurihara T, Jiang X, et al. Scleral PERK and ATF6 as targets of myopic axial elongation of mouse eyes. Nat Commun. 2022;13(1):5859. Published 2022 Oct 10. doi:10.1038/s41467-022-33605-1
[2] Kang L, Ikeda SI, Yang Y, et al. Establishment of a novel ER-stress induced myopia model in mice. Eye Vis (Lond). 2023;10(1):44. Published 2023 Nov 1. doi:10.1186/s40662-023-00361-2
11- In discussion, avoid overstatement (e.g., line 229: "Ultimately disrupts" implies causality without direct proof). Compare with human studies more (e.g., ref. 16 on PS-OCT in high myopia).
Response:
We appreciate the reviewer’s valuable comment.
We agree that the original phrase “ultimately disrupts” could imply a direct causal relationship. In the revised Discussion, this expression has been softened to “may compromise” to better reflect the observational nature of our findings.
In addition, we have incorporated a comparison with human PS-OCT studies in highly myopic eyes[1], which reported abnormalities in scleral fiber orientation and birefringence.
This addition highlights the translational relevance of our model and indicates that the collagen disorganization observed in mice is in line with the structural abnormalities detected in human high myopia.
Page 10: line 249-252:
Human PS-OCT study has reported reduced scleral birefringence in highly myopic eyes, which likely reflects decreased collagen packing and reduced structural anisotropy due to disorganization of collagen fibers.
[1] Ohno-Matsui, K.; Igarashi-Yokoi, T.; Azuma, T.; Sugisawa, K.; Xiong, J.; Takahashi, T.; Uramoto, K.; Kamoi, K.; Okamoto, M.; Banerjee, S.; et al. Polarization-Sensitive OCT Imaging of Scleral Abnormalities in Eyes With High Myopia and Dome-Shaped Macula. JAMA Ophthalmol. 2024, 142, 310-319, doi:10.1001/jamaophthalmol.2024.0002.
12- The limitation of 2D imaging is acknowledged, but potential solutions (e.g., PLM tomography or second-harmonic generation microscopy) could be mentioned.
Response:
We thank the reviewer for this constructive suggestion.
In the revised Discussion, we have added a note on advanced optical approaches that could help overcome the limitation of two-dimensional imaging. These techniques represent promising directions for our future work. We sincerely appreciate the reviewer’s valuable guidance.
Page 11: line 308-310:
Future studies using polarization-resolved second-harmonic generation microscopy may provide true three-dimensional insights into collagen organization beyond the two-dimensional limitations of PLM.
13- Figures 5C–D present mean ± SD values but lack individual data points, making it difficult to judge variability. Include scatter plots of individual animal data. Report effect size and confidence intervals, not just p-values, to better represent statistical reliability.
Response:
We thank the reviewer for this valuable comment.
In the revised manuscript, individual data points for each animal have been added to Figure 6 (as below, previously Figure 5) to more clearly illustrate within-group variability. In addition, effect sizes (Cohen’s dz) and 95% confidence intervals have been reported in the text to provide a more comprehensive representation of statistical reliability.

Page 8: line 193-196:
The eyes treated with −30 D lenses exhibited a significant myopic shift in refraction (−30 D vs. no lens: -8.42 ± 3.87D vs. 4.42 ± 4.45 D; P = 0.002, dz = 2.60, 95% CI: 6.70–18.97) and axial length elongation (−30 D vs. no lens: 0.21 ± 0.04 mm vs. 0.18 ± 0.05 mm; P = 0.020, dz = 1.34, 95% CI: -0.05 to 0.00) compared to the contralateral controls.
Page 8: line 202-206:
Quantitative analysis revealed a significant decrease in both the proportion (−30 D vs. no lens: 40.91 ± 6.58% vs. 48.36 ± 6.47%, P = 0.040, dz = 1.04, 95% CI: -0.01 to 0.16) and average size (−30 D vs. no lens: 147.11 ± 59.38µm2 vs. 281.45 ± 101.00 µm2, P = 0.002, dz = 2.65, 95% CI: 71.36 to 197.30) of collagen fiber bundles in myopic eyes compared with the contralateral controls.
Reviewer 3 Report
Comments and Suggestions for Authors
This manuscript aims to investigate polarized light microscopy-based quantification of scleral collagen fiber bundle remodeling in the lens-induced myopia mouse model. Although the topic is interesting in its scientific field, there are some issues that require the authors’ attention to improve the quality of this particular manuscript before further consideration for publication in a high-quality journal “Life”.
Specific comments:
- Why the authors select the myopic conditions of -30D and a 3-week induction for evaluation? Please justify.
- Furthermore, the endpoint of experimental induction is week 3. Nevertheless, in my opinion, the time-course data are also important to understand the myopia-mediated process of decreasing collagen bundle ratio and the bundle area. If possible, please also include the experimental evidences at time points of week 1 and 2.
- What are the instruments used to measure refraction and axial length (model, wavelength, and scanning mode)? Furthermore, the refractive changes before and after anesthesia, and corneal curvature settings and correction methods are unclear to the audiences? Please specify.
- What are the major causing factors for the thinning of collagen bundles? Are the outcomes correlated with the degradation or reorientation of collagenous matrix? Please justify.
- Why all the mice used in this study are male? Whether any gender factor is under consideration for the experimental design? Please clarify.
- As mentioned in Section 2.3, quantitative image analysis was performed using ImageJ software. Although this is a routine experimental claim, many investigators indeed adopt the powerful tool of ImageJ software to conduct quantitative image analysis of ultrastructure of eye tissues (please see Figure S12 of the article DOI: 10.1002/advs.202302174). If possible, please consider the inclusion of the aforementioned relevant case study in the reference list to support the methodology and attract more attention from broad readers.
Author Response
Comment 1: Why the authors select the myopic conditions of -30D and a 3-week induction for evaluation? Please justify.
Response:
We appreciate the reviewer’s insightful question.
During the development of this model[1], our laboratory systematically compared different lens powers (–10D, –20D, and –30D). The –30D produced the most pronounced myopic changes in both refraction and axial length, and therefore was adopted as the standard condition for subsequent experiments.
Regarding the duration, a 3-week induction period was chosen to cover the critical developmental window from 3 to 6 weeks of age, during which the mouse eye undergoes rapid growth and the sclera exhibits high sensitivity to visual defocus. This period allows effective detection of early tissue changes associated with myopia induction.
In our subsequent long-term experiments, we observed that after 6 weeks of age, ocular growth markedly slows and myopia progression becomes less pronounced[2]. These findings further support that the 3- to 6-week window is optimal for investigating early scleral remodeling in lens-induced myopia.
[1] Jiang X, Kurihara T, Kunimi H, et al. A highly efficient murine model of experimental myopia. Sci Rep. 2018;8(1):2026. Published 2018 Feb 1. doi:10.1038/s41598-018-20272-w
[2] Yang Y, Tomita Y, Lee D, et al. Impact of Extended Lens-Induced Myopia on Retinal Structure and Function in Mice. Curr Eye Res. Published online September 16, 2025. doi:10.1080/02713683.2025.2557590
Comment 2: Furthermore, the endpoint of experimental induction is week 3. Nevertheless, in my opinion, the time-course data are also important to understand the myopia-mediated process of decreasing collagen bundle ratio and the bundle area. If possible, please also include the experimental evidences at time points of week 1 and 2.
Response:
We sincerely thank the reviewer for this valuable suggestion.
The primary aim of this study was to establish and validate a quantitative method for scleral collagen bundle analysis using polarized light microscopy; therefore, the 3-week induction point was selected as a standard endpoint for morphological assessment.
We fully agree that time-course data would provide additional insights into the dynamic nature of collagen remodeling during myopia progression. Although additional experiments at earlier time points (weeks 1 and 2) could not be completed within the current study due to the experimental cycle, we have planned to incorporate these time-course analyses in our ongoing and future work to further expand the understanding of temporal remodeling patterns.
Comment 3: What are the instruments used to measure refraction and axial length (model, wavelength, and scanning mode)? Furthermore, the refractive changes before and after anesthesia, and corneal curvature settings and correction methods are unclear to the audiences? Please specify.
Response:
We thank the reviewer for the detailed comments.
In the revised manuscript, we have added the instrument information and operational details for both refraction and axial length measurements.
Refraction was measured under general anesthesia. Although anesthesia may slightly influence refractive state, we previously attempted measurements in awake mice and found unstable readings and poor reproducibility (Response Figure 1)[1]. Therefore, refractive assessments were conducted under anesthesia to ensure data consistency and reliability.
In our earlier investigations, we also examined corneal curvature changes during myopia development and found no significant difference between myopic and control eyes (Response Figure 2)[1]. Accordingly, corneal curvature was not included as a parameter in the present study.

Response Figure 1. Comparison of infrared photorefraction in mice under awake (top) and anesthetized (bottom) conditions, reproduced from our earlier publication (Adapted from Figure 2b.c in reference [1] ).

Response Figure 2. no significant difference between myopic and control eyes (Adapted from Supplementary Figure 10 in reference [1] ).
Page 3: line 99-102:
Measurements of refraction and axial length were performed at postnatal week 3 and week 6 using an infrared photorefractor (Steinbeis Transfer Center, Plochingen, Germany) and spectral-domain optical coherence tomography (SD-OCT; Envisu R4310, Leica Microsystems, Wetzlar, Germany), respectively. .
[1] Jiang X, Kurihara T, Kunimi H, et al. A highly efficient murine model of experimental myopia. Sci Rep. 2018;8(1):2026. Published 2018 Feb 1. doi:10.1038/s41598-018-20272-w
Comment 4: What are the major causing factors for the thinning of collagen bundles? Are the outcomes correlated with the degradation or reorientation of collagenous matrix? Please justify.
Response:
We thank the reviewer for this insightful comment.
To clarify, the Discussion section describes that the thinning of collagen bundles results from a multi-level remodeling process involving both degradation and reorganization of the collagenous matrix. Specifically, ER stress impairs collagen synthesis and assembly, while MMP-2–mediated ECM degradation and lamellar slippage contribute to the loss of bundle integrity.
We have slightly revised the text to make this causal explanation clearer in the revised manuscript.
Page 11: line 281-290:
Our previous studies have demonstrated that endoplasmic reticulum (ER) stress is activated during myopia progression, which interferes with the synthesis and assembly of collagen, resulting in thinner individual collagen fibers and a reduction in total collagen content. In addition, collagen fibers are embedded within a proteoglycan rich, gel-like matrix that resists interlamellar shearing and stabilizes the lamellar alignment. During myopia progression, matrix metalloproteinase-2 (MMP-2), a key enzyme upregulated in scleral remodeling, promotes degradation of the extracellular matrix. With progressive matrix weakening, collagen lamellae lose structural support and begin to slide relative to one another, leading to scleral creep and axial elongation.
Comment 5: Why all the mice used in this study are male? Whether any gender factor is under consideration for the experimental design? Please clarify.
Response:
We sincerely thank the reviewer for raising this valuable and scientifically important point.
Sex differences represent a critical but often underappreciated factor in myopia research. In accordance with the reviewer’s suggestion, we have added a discussion on this issue in the revised Discussion section.
Worldwide, most experimental myopia models have been established using male animals. This choice is primarily driven by scientific and experimental control considerations. Female mice undergo regular estrous cycles characterized by marked fluctuations in estrogen and progesterone levels. Such hormonal variability can increase data noise and experimental uncertainty, thereby reducing reproducibility. In contrast, male mice show more linear and stable ocular growth patterns, providing statistical advantages for standardized model development.
Nevertheless, we fully agree that using only males is a limitation. Previous studies have suggested that sex hormones may regulate scleral ECM metabolism[1], and therefore, exclusive use of males could mask sex-specific differences.
In our own laboratory, we performed lens-induced myopia (LIM) experiments in both male and female mice starting at 3 weeks of age. However, the female mice did not show a statistically significant axial elongation after three weeks of induction (Response Figure1)[2], indicating a lower sensitivity to lens-induced myopic stimulus under the same conditions. For this reason, male mice were selected in the present study to ensure model stability and reproducibility.
We are aware that an increasing number of studies are now reintroducing female models to explore sex-dependent mechanisms. Our group is also actively optimizing conditions to establish a stable female model. This direction is clinically meaningful, as large epidemiological studies have shown that adolescent females tend to develop myopia earlier and at a higher prevalence than males[3]. Thus, incorporating female animals in future experiments would help to better simulate sex-related differences in myopia progression, scleral remodeling, and treatment responses, thereby improving the translational relevance of experimental myopia research.

Response Figure 1. Changes in axial length in female mice after 3 weeks of −30 D lens-induced myopia. No significant differences were observed between the -30D and control eyes. (Adapted from Figure 1 in reference [2].)
Page 11: line 294-302:
It should be noted that nearly all existing experimental myopia models have been developed using male animals, including the present study. This is primarily due to methodological considerations, male mice display stable ocular growth trajectories, avoiding variability introduced by the estrous cycle. Consequently, male models have become the global standard for establishing reproducible myopia paradigms. Nevertheless, accumulating evidence suggests that sex hormones may modulate scleral ECM metabolism and influence susceptibility to myopia. Investigating female models under controlled conditions will be important for elucidating sex-dependent mechanisms and improving the translational relevance of experimental myopia research.
[1] Huang L, Zhang D, Zhou J. Myopia development: multifactorial interplay, molecular mechanisms and possible strategies. Front Med (Lausanne). 2025;12:1638184. Published 2025 Aug 26. doi:10.3389/fmed.2025.1638184
[2] Zhang Y, Mori K, Jeong H, et al. Myopic shift in female mice after ovariectomy. Sci Rep. 2024;14(1):22946. Published 2024 Oct 3. doi:10.1038/s41598-024-74337-0
[3] Machluf Y, Israeli A, Cohen E, Chaiter Y, Mezer E. Dissecting the complex sex-based associations of myopia with height and weight. Eye (Lond). 2024;38(8):1485-1495. doi:10.1038/s41433-024-02931-7
Comment 6: As mentioned in Section 2.3, quantitative image analysis was performed using ImageJ software. Although this is a routine experimental claim, many investigators indeed adopt the powerful tool of ImageJ software to conduct quantitative image analysis of ultrastructure of eye tissues (please see Figure S12 of the article DOI: 10.1002/advs.202302174). If possible, please consider the inclusion of the aforementioned relevant case study in the reference list to support the methodology and attract more attention from broad readers.
Response:
We appreciate the reviewer’s constructive suggestion.
In the revised manuscript, we have added the recommended reference to support the use of ImageJ software for quantitative analysis of ocular ultrastructure.
This citation helps to strengthen the methodological rationale and improve the contextual relevance of our study.
Round 2
Reviewer 2 Report
Comments and Suggestions for Authors
Dear Editor
Thank you for providing me the opportunity to review this manuscript
The authors have thoroughly addressed all my concerns. The revisions are comprehensive and responsive, aligning well with the comments. The manuscript is now stronger, with enhanced scientific and editorial quality.
I have no further comments; therefore, I recommend the publication of the manuscript.
Best wishes
Reviewer 3 Report
Comments and Suggestions for Authors
The revised version has adequately addressed most of the critiques raised by this reviewer and is now suitable for publication in "Life".